# A Novel 3-Gene Signature for Identifying COVID-19 Patients Based on Bioinformatics and Machine Learning

**DOI:** 10.3390/genes13091602

**Published:** 2022-09-08

**Authors:** Guichuan Lai, Hui Liu, Jielian Deng, Kangjie Li, Biao Xie

**Affiliations:** Department of Epidemiology and Health Statistics, School of Public Health, Chongqing Medical University, Yixue Road, Chongqing 400016, China

**Keywords:** coronavirus disease 2019, biomarker, CIBERSORT, WGCNA, GO, LASSO, Boruta, multivariate logistic regression

## Abstract

Although many biomarkers associated with coronavirus disease 2019 (COVID-19) were found, a novel signature relevant to immune cells has not been developed. In this work, the “CIBERSORT” algorithm was used to assess the fraction of immune infiltrating cells in GSE152641 and GSE171110. Key modules associated with important immune cells were selected by the “WGCNA” package. The “GO” enrichment analysis was used to reveal the biological function associated with COVID-19. The “Boruta” algorithm was used to screen candidate genes, and the “LASSO” algorithm was used for collinearity reduction. A novel gene signature was developed based on multivariate logistic regression analysis. Subsequently, M0 macrophages (PR_AUC_ = 0.948 in GSE152641 and PR_AUC_ = 0.981 in GSE171110) and neutrophils (PR_AUC_ = 0.892 in GSE152641 and PR_AUC_ = 0.960 in GSE171110) were considered as important immune cells. Forty-three intersected genes from two modules were selected, which mainly participated in some immune-related activities. Finally, a three-gene signature comprising CLEC4D, DUSP13, and UNC5A that can accurately distinguish COVID-19 patients and healthy controls in three datasets was constructed. The ROC_AUC_ was 0.974 in the training set, 0.946 in the internal test set, and 0.709 in the external test set. In conclusion, we constructed a three-gene signature to identify COVID-19, and CLEC4D, DUSP13, and UNC5A may be potential biomarkers for COVID-19 patients.

## 1. Introduction

Infectious disease caused by coronavirus SARS-CoV-2 infection is known as coronavirus disease 2019 (COVID-19) [1]. At the time of writing, there were more than 500 million confirmed cases of COVID-19, including six million deaths (https://covid19.who.int, accessed on 15 June 2022). Although most COVID-19 vaccines can successfully protect against the COVID-19 virus, individual heterogeneity in immune systems has shaped the effectiveness of vaccines [2]. The adaptive immune response is a major determinant affecting virus clearance and the adoption of vaccines after SARS-CoV-2 infection [3]. The adaptive immune system mainly includes B cells, CD4^+^ T cells, and CD8^+^ T cells, and they play a vital protective role in combating viral infections [4]. The generation of SARS-CoV-2-specific memory B cells could provide persistent protection against repeat infection [5]. SARS-CoV-2-specific CD4^+^ T cells can improve the ability of SARS-CoV-2 clearance [6]. A potential protective role of CD8^+^ T cell responses in mild COVID-19 patients was detected because of a higher fraction of CD8^+^ T cell responses observed at this stage [7]. These findings can emphasize the significance of SARS-CoV-2-specific immune cells in killing the virus.

The differential characteristic associated with immune cells between COVID-19 patients, and non-COVID-19 patients or healthy controls has been investigated in many studies [8,9,10]. Wauters et al. found that COVID-19 patients had a higher immune infiltrating of neutrophil subclusters compared to those with non-COVID-19, but lower CD4^+^ T-helper-17 cells appeared in COVID-19 patients [8]. Jing et al. found significantly elevated levels of HK2, SLC04A1, SDS, and COL1A1 in B cells in non-COVID-19 patients compared with those with COVID-19 infections [9]. Rébillard et al. found that a decreased proportion of T cells was correlated with the cause of acute SARS-CoV-2 infection [10]. Although many biomarkers identifying COVID-19 have been developed, a novel gene signature which focused on the immune microenvironment related to COVID-19 has not been constructed. With the development of new technology, many studies have focused on combining machine learning (ML) and bioinformatics for identifying and predicting the outcome of COVID-19 (Appendix A) [11,12,13,14]. Although these studies focused on predictive biomarkers associated with COVID-19 using bioinformatics and ML, these biomarkers were only validated in a single dataset without other external test sets. Meanwhile, the immune microenvironment related to COVID-19 should be explored because of its impact on the development of COVID-19 and patients’ responses to vaccines. Therefore, we aimed to identify a predictive biomarker associated with immune cells infected by COVID-19 based on ML and bioinformatics, followed by validation in two test sets in order to ensure the stability of the results.

In the present work, we used bioinformatics analyses to determine important immune cells and key modules associated with these immune cells. ML analyses were used to screen and construct a novel gene signature that can effectively distinguish COVID-19 patients and normal controls based on these candidate genes from these key modules.

## 2. Materials and Methods

### 2.1. Data Acquisition

In this work, we recruited 86 American samples consisting of 62 COVID-19 patients and 24 healthy controls in GSE152641 (http://www.ncbi.nlm.nih.gov/geo/, accessed on 1 June 2022); 54 French samples comprising 44 severe COVID-19 patients and 10 healthy donors in GSE171110 (http://www.ncbi.nlm.nih.gov/geo/, accessed on 1 June 2022); and 49 Indian samples composed of 8 asymptomatic COVID-19 patients, 9 mild COVID-19 patients, 10 moderate COVID-19 patients, 7 severe COVID-19 patients, 6 COVID-19 bacterially infected patients, and 9 healthy controls in GSE196822 (http://www.ncbi.nlm.nih.gov/geo/, accessed on 24 August 2022). Transcriptomic data from the three datasets were all extracted from their whole blood and developed on different platforms, which were involved in GPL24676 (Illumina NovaSeq 6000), GPL16791 (Illumina HiSeq 2500), and GPL20301(Illumina HiSeq 4000). All mRNA expression profiles were displayed with count-based data, and we subsequently transformed them into transcripts per million. The enrolled criteria were shown as the following: 1. COVID-19 patients were clinically diagnosed as COVID-19 without other infectious diseases, and healthy controls were people from normal populations without COVID-19 or other infectious diseases. 2. All samples with complete mRNA and outcomes were considered. Finally, 62 COVID-19 patients and 24 healthy controls from GSE152641, 44 COVID-19 patients and ten healthy donors from GSE171110, and 34 COVID-19 patients and nine healthy controls from GSE196822 were enrolled into this work. The work flow is shown in Figure 1.

### 2.2. Calculation of Immune Cell

We used the “CIBERSORT” (https://cibersort.stanford.edu/, accessed on 5 June 2022) deconvolution algorithm to quantify 22 immune cells. We ran the CIBERSORT algorithm with 1000 permutations based on normalized gene expression profiles and LM22, including a gene expression matrix of 22 immune cells [15]. Then, samples with *p* < 0.05 were included in this work. We compared the proportion of immune cells between COVID-19 patients and healthy controls. The area under the curve (AUC) under precision–recall (PR) curves calculated via the “modEvA” package is used to identify the predictive abilities of 22 immune cells. Finally, immune cells with differential distribution and high predictive power were determined as the important immune cells.

### 2.3. Differentially Expressed Analysis

We performed the differentially expressed analysis between the COVID-19 patients and healthy controls with the “edgeR” package, which is usually used for count-based expression data [16]. Genes with |log_2_FC| > 1 and FDR < 0.05 were regarded as differentially expressed genes.

### 2.4. WGCNA Analysis

The DEGs were used to construct a weight co-expression network via the “WGCNA” package [17]. The β when scale-free topology fitting index R^2^ > 0.85 is in accordance with the optimal β calculated by “powerEstimate”. Thus, we chose an optimal soft thresholding power (β) according to scale-free topology fitting index R^2^ > 0.85. We acquired the adjacency matrix based on the gene expression matrix and the optimal power. The adjacency matrix was converted into the topological overlap measure (TOM) matrix. TOM is used to reflect the similarity of co-expression genes. We performed the clustering analysis with average linkage hierarchical clustering dependent on the dissimilarity of TOM. We obtained genes with high similarity in their co-expression in the same module (minimum size = 30) using the “dynamic tree cutting” algorithm. We used the “Merged dynamic” algorithm to ensure these modules with a high correlation degree using a cutHeight = 0.2 as the cutoff value. We used the fractions of the important immune-infiltrating cells of COVID-19 patients as sample traits. The important modules were identified according to a high correlation between these co-expression gene modules and sample traits. Additionally, the genes in the important modules were the key genes and the intersected key genes between GSE152641 and GSE171110 were included as candidate genes in our work.

### 2.5. GO Enrichment Analysis

To determine the biological significance of these candidate genes in the development of COVID-19, gene ontology (GO) enrichment analysis was employed based on “org.Hs.eg.db”, “ggplot2”, and “clusterprofiler” packages [18]. The GO enrichment analysis comprised cellular components, biological properties, and molecular functions. The biological functions with adjusted *p* < 0.05 were visualized.

### 2.6. Construction and Validation of a Novel Gene Signature

In GSE152641, the Boruta algorithm with 500 maxRuns was applied to further screen important genes among candidate genes using the “Boruta” package. Genes were divided into three categories including confirmed, tentative, and rejected importance. We put genes with confirmed importance into the least absolute shrinkage and selection operator (LASSO), elastic net regression, and ridge regression analyses. The minimum lambda value was obtained from ten-fold cross-validation. We divided the GSE152641 dataset into training and testing datasets with a 1:1 ratio. Next, we used a “glmnet” package to perform LASSO, ridge, and elastic net regressions. The mean square error (MSE) reflects the predictive power of models. A lower MSE indicates a higher predictive value. Genes with nonzero regression coefficients were employed in a multivariate logistic regression analysis. We constructed a score based on genes with *p* < 0.05 in the model. Score = Constant + Gene_1_ × Coef_1_ + Gene_2_ × Coef_2_ + Gene_3_ × Coef_3_ + ⋯⋯Gene_n_ × Coef_n_. Constant, Gene, and Coef represent the constant, gene expression, and regression coefficient, respectively. The AUC under the receiver-operating characteristic curve (ROC) is used to test the predictive power by a “pROC” package. GSE171110 was used to validate the stability of this signature. Additionally, we used GSE196822 to validate two important immune cells and the applicability of this signature.

### 2.7. Statistical Analysis

All analyses in this work were based on R 4.0.3. The PR_AUC_ is used to compare the predictive power of immune cells. The Wilcoxon analysis is used to analyze the differences between the two groups, including the comparison of 22 immune cells and candidate signatures between COVID-19 patients and normal controls. The ROC_AUC_ is used to assess the predictive power of a gene signature. A two-sided *p* value < 0.05 was considered statistically significant.

## 3. Results

### 3.1. Identification of Two Important Immune Cells

In GSE152641, we found that healthy controls had a higher immune infiltrating level of naïve B cells, activated NK cells, resting memory CD4^+^ T cells, naïve CD4^+^ T cells, and CD8^+^ T cells compared to those in COVID-19 patients, while higher proportions of M0 macrophages, neutrophils, plasma cells, and γ-delta T cells were found in COVID-19 patients than those in healthy controls (Figure 2A). In GSE171110, the fraction of resting dendritic cells, M2 macrophages, resting NK cells, resting memory CD4^+^ T cells, and CD8^+^ T cells were significantly higher in the healthy controls than those in the COVID-19 patients, but lower proportions of M0 macrophages, neutrophils, and plasma cells were observed in the healthy controls compared with COVID-19 patients(Figure 2B). In addition, the AUC of these differentially distributed immune cells in the prediction of outcomes showed that M0 macrophages (Figure 3A,K) and neutrophils (Figure 3B,L) had higher predictive values. Therefore, the two immune cells were considered important immune cells in this work.

### 3.2. Determination of Key Modules

In this work, 1997 up-regulated and 183 down-regulated DEGs were visualized by volcano plot in GSE152641 (Figure 4A), and 2150 up-regulated and 752 down-regulated DEGs were visualized by volcano plot in GSE171110 (Figure 4B). Next, we determined the key module associated with two important immune cells in two datasets. In GSE152641, we chose β = 7 as the optimal β and acquired ten modules using the “merged dynamic” algorithm (Figure 5A,C). We found that the brown module was highly correlated with the M0 macrophages (cor = 0.62, *p* = 6 × 10^−8^) and neutrophil (cor = 0.80, *p* = 4 × 10^−15^) in GSE152641 (Figure 5E). Similarly, we chose β = 14 as the optimal β and acquired 12 modules using the “merged dynamic” algorithm in GSE171110 (Figure 5B,D). In addition, the green module was highly correlated with M0 macrophages (cor = 0.69, *p* = 3 × 10^−7^) and neutrophil (cor = 0.85, *p* = 2 × 10^−13^) in GSE171110 (Figure 5F). Therefore, the brown module in GSE152641 and the green module in GSE171110 were selected as key modules. We intersected the genes between brown and green modules and ultimately obtained 43 candidate genes (Figure 6A). The results of GO enrichment analysis showed that these candidate genes participated in some immune-related activities, including neutrophil degranulation, neutrophil activation involved in immune response, fatty acid binding, RAGE receptor binding, and Toll-like receptor (TLR) binding (Figure 6B).

### 3.3. Construction and Validation of a 3-Gene Signature

In GSE152641, 43 candidate genes were divided into three categories including 19 confirmed genes, 3 tentative genes, and 21 rejected genes. After a comparison of three algorithms (LASSO, ridge, and elastic net regressions), the lowest MSE appeared in the LASSO regression model, while the highest MSE was found in the ridge regression model (Appendix A). Therefore, we chose the LASSO as the selective method. Next, 19 genes with confirmed importance were applied in the LASSO analysis, and we further selected seven genes with nonzero regression coefficients according to lamda = 0.01831 (Figure 6C,D). The detailed results of the Boruta and LASSO algorithms are displayed in Table 1. Finally, a novel gene signature was constructed based on the stepwise multivariate logistic regression analysis. Score = −23.3719 + 2.5963 × CLEC4D + 2.3694 × DUSP13 + 2.0826 × UNC5A. Samples with higher scores tended to have a higher probability of COVID-19, and significantly higher expressions of CLEC4D, DUSP13, and UNC5A were found in COVID-19 patients compared with those in healthy controls in GSE152641 (Figure 7A,C–E) and GSE171110 (Figure 8A,C–E). Most importantly, the AUC was 0.974(0.944–1) and 0.946(0.885–1) in GSE152641 (Figure 7B) and GSE171110 (Figure 8B), respectively, indicating a high value of our signature in predicting the outcome.

### 3.4. External Validation of Candidate Immune Cells and Genes in GSE196822

To test the applicable value of this signature, we used another external dataset, GSE196822, to ensure the stability of these results. We found higher immune infiltrations of M0 macrophages and neutrophils and significantly elevated levels of DUSP13 and UNC5A in COVID-19 (Appendix A). Although no difference in CLEC4D was found between COVID-19 patients and healthy controls, the differentially expressed analysis demonstrated that CLEC4D was an up-regulated gene (log_2_FC = 1.808, FDR = 0.0306) in COVID-19 infections (Appendix A). Overall, the AUC in GSE196822 was 0.709, showing a predictive power of our signature (Appendix A). Finally, these findings further confirmed the results above.

## 4. Discussion

Many SARS-CoV-2-specific immune cells have been proven to have a crucial role in cleaning or killing the COVID-19 virus and shaping the efficacy of COVID-19 vaccines. Therefore, constructing a novel SARS-CoV-2-related gene signature associated with immune cells is important for targeted therapy for COVID-19 patients in the future.

In this work, we acquired the proportion of 22 immune cells using the “CIBERSORT” algorithm. Meanwhile, we performed a differential analysis on immune cells between COVID-19 patients and healthy controls and found that two of these differentially distributed immune cells had a higher predictive value and thus were defined as the important immune cells. An increased fraction of M0 macrophages and neutrophil were enriched in the COVID-19 patients compared to healthy controls, which was consistent with a previous study [19]. Inflammation augments and immune-related pathways (e.g., TLR signaling pathways) in macrophages were activated through SARS-CoV-2 infection [20]. The overexpression of inflammatory factors in M0 macrophages increased the severity of COVID-19 [20]. Therefore, the M0 macrophages were related to the progression of COVID-19 through activated inflammation induced by COVID-19. Neutrophils contributed to the development of COVID-19, which was associated with the severity of COVID-19 [21]. Additionally, neutrophil was a symbol that differentiated the non-severe from severe COVID-19 patients [22]. Moderately suppressing the boast of neutrophils was the main therapeutic strategy for reducing the probability of continual infection caused by COVID-19 [23,24]. These findings above further confirmed that M0 macrophages and neutrophils played an important role in the development of COVID-19.

Forty-three candidate genes from key modules that were highly correlated with two important immune cells were involved in immune-related activities. Additionally, most of these biological functions were associated with the development of the COVID-19 virus. For example, severe COVID-19 patients showed high expressions of genes involved in neutrophil degranulation, and increased degranulation of neutrophils affected the immune response of COVID-19 patients [25]. Bankar et al. also found that neutrophil degranulation was one of the major manifestations that regulated the immune system after COVID-19 infection [26]. Inflammatory response contributed to lung injury for COVID-19 patients and an unfavorable inflammatory status was a sign of more severe disease [27,28]. Inflammatory response to SARS-CoV-2 infection was critical to appropriate therapy for COVID-19 patients [29]. RAGE receptor binding induced inflammatory responses, which may be a potential target for inflammatory disease during SARS-CoV-2 and a biomarker related to viral infection, including COVID-19 [30,31]. The level of intestinal fatty acid binding proteins was higher in COVID-19 patients than those in healthy donors [32]. The hyperstimulation of TLR signaling can initiate immunopathology after infection with viruses, allowing the monocytes from recovered patients to produce anti-viral responses [33]. However, the inability of immune response caused by TLR tolerance was associated with COVID-19, leading to a worse outcome [33]. Most importantly, neutrophils are a crucial immune cell associated with COVID-19, and the functions of the 43 candidate genes involved were associated with most neutrophil-related activities. Therefore, we thought that these neutrophil-related genes may become potential targets for COVID-19. To sum up, 43 candidate genes may be involved in the development of COVID-19 through immune-related functions.

Finally, we used ML approaches to accurately obtain the key genes in order to construct a predictive gene signature. The Boruta algorithm is often used for dimensionality reduction, and the LASSO algorithm is used for collinearity reduction. A novel gene signature consisting of three genes (CLEC4D, DUSP13, and UNC5A) was identified through multivariate logistic regression analysis based on the gene expression profile in GSE152641. The expressions of the three genes were all higher in COVID-19 patients than those observed in healthy controls. In the previous study, the CLEC4 family played a critical role in immune response and the development of hepatocellular carcinoma. However, CLEC4D, as one of the CLEC4s, demonstrated a potential value in virus infection, including COVID-19 [34]. DUSP13 was associated with the regulation of the MAPK signaling pathway after stress stimulation in cardiomyocytes [35]. The up-regulation of DUSP13 to cardiac stress contributed to coronary artery disease, a common cardiovascular comorbidity of COVID-19 [36,37]. UNC5A retrained virus replication through autophagy induction instead of apoptosis [38]. Recently, Plissonnier et al. found that UNC5A expression was obviously decreased in clinical Hepatitis C virus (+) specimens compared to that of uninfected samples [39]. The present model achieved a high AUC in the training and test sets. It was noteworthy that COVID-19 patients displayed higher scores than those of healthy controls. Therefore, these findings can demonstrate that our gene signature is an excellent biomarker for identifying COVID-19 patients.

There were several inspiring insights from previous studies, which motivated the current research. Firstly, as we know, a comprehensive ML includes regression, classification, clustering, and dimensionality reduction [40]. Additionally, molecular subtypes contribute to the treatment of COVID-19 [41,42,43]. Therefore, we should consider the application of ML in the classification of more COVID-19 patients based on some immune-related traits to improve the immune response to COVID-19 vaccines in the future. Secondly, the integration of deep learning and medical images has improved COVID-19 diagnosis and prediction [44,45]. Next, we should fully combine the genomic data and medical images when utilizing deep learning to identify COVID-19 patients. Finally, with the development of ML in the design of potential drugs and vaccines [45], we should further focus on improving effective vaccine discovery and identifying potential COVID-19 patients benefiting from vaccines.

## 5. Conclusions

In conclusion, our work reveals the relationship between immune cells and COVID-19 further determined by two important immune cells. At the same time, we discovered that 43 genes were correlated with these important immune cells. Most importantly, various approaches were utilized to construct a novel gene signature, which had a high predictive value in various datasets. Therefore, the key contributions of this work are that we developed a novel signature that can accurately identify COVID-19 patients from different platforms and solved the problem of little attention being paid to genes relevant to the immune cells of COVID-19. These contributions can strengthen the potential role of two important immune cells and three candidate genes in targeting COVID-19. However, there were several limitations in this work. Firstly, all data and obtained results were based on public databases; further experimental analysis is needed to confirm these findings. Secondly, although a high-predictive power of this signature was found in this work, the stability of this constructed model depended on the current sample size. Finally, the lack of clinical information on COVID-19 leads to a difficult combination with our novel signature in comprehensively identifying COVID-19 patients.

Lastly, future scopes should include three goals: Firstly, future researches should focus on predicting severity in COVID-19 and classifying COVID-19 to decrease the medical cost and to improve the efficacy of vaccines. Secondly, we should also integrate genomic data, medical images, and deep learning to determine an appropriate treatment plan, improving COVID-19 patients’ overall immune response to vaccines. Finally, although we have discovered three candidate genes that help identify COVID-19 in this work, how some mechanisms and functions of the three genes impact COVID-19 should be explored in the future.

## Figures and Tables

**Figure 1 genes-13-01602-f001:**
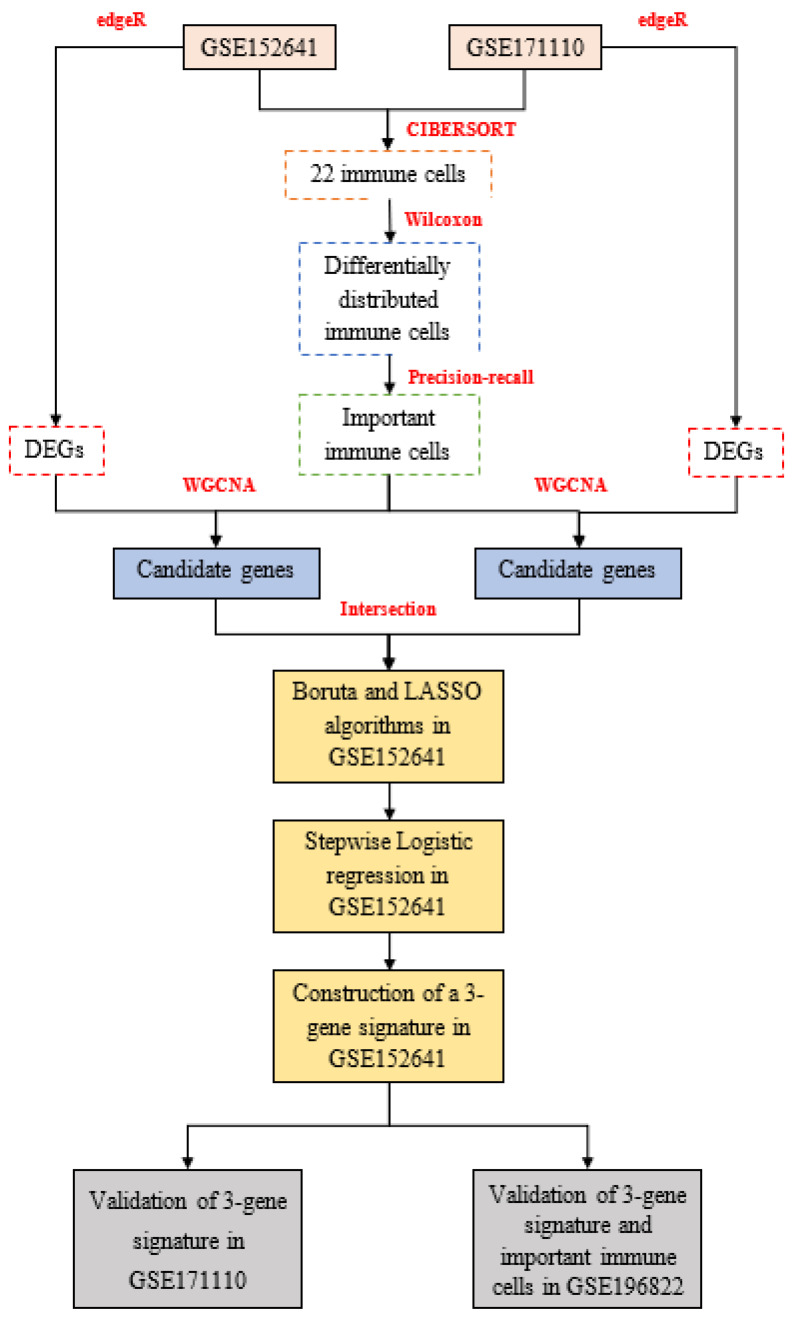
The workflow of this study: **edgeR:** empirical analysis of digital gene expression in R, **CIBERSORT:** cell-type identification by estimating relative subsets of RNA transcripts, **DEGs:** differentially expressed genes, **WGCNA:** weighted gene co-expression network analysis, **LASSO:** least absolute shrinkage and selection operator.

**Figure 2 genes-13-01602-f002:**
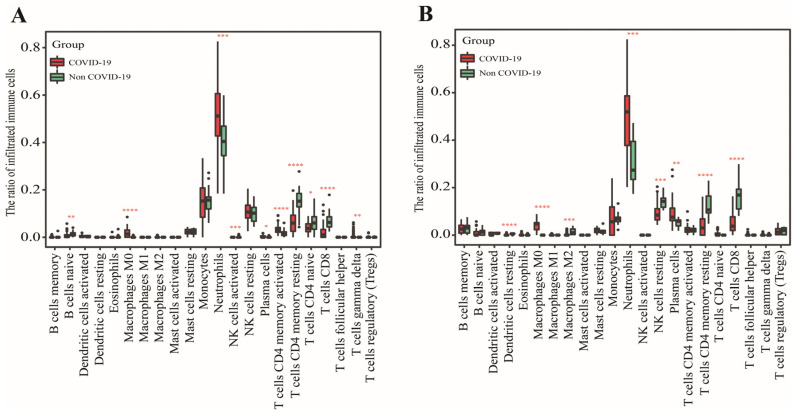
The determination of differentially distributed immune cells. (**A**) The fraction of 22 immune cells between COVID-19 patients and healthy controls in GSE152641. (**B**) The fraction of 22 immune cells between COVID-19 patients and healthy controls in GSE171110. Data were analyzed by Wilcoxon test; ns, no significance; * *p* < 0.05, ** *p* < 0.01, *** *p* < 0.001 and **** *p* < 0.0001.

**Figure 3 genes-13-01602-f003:**
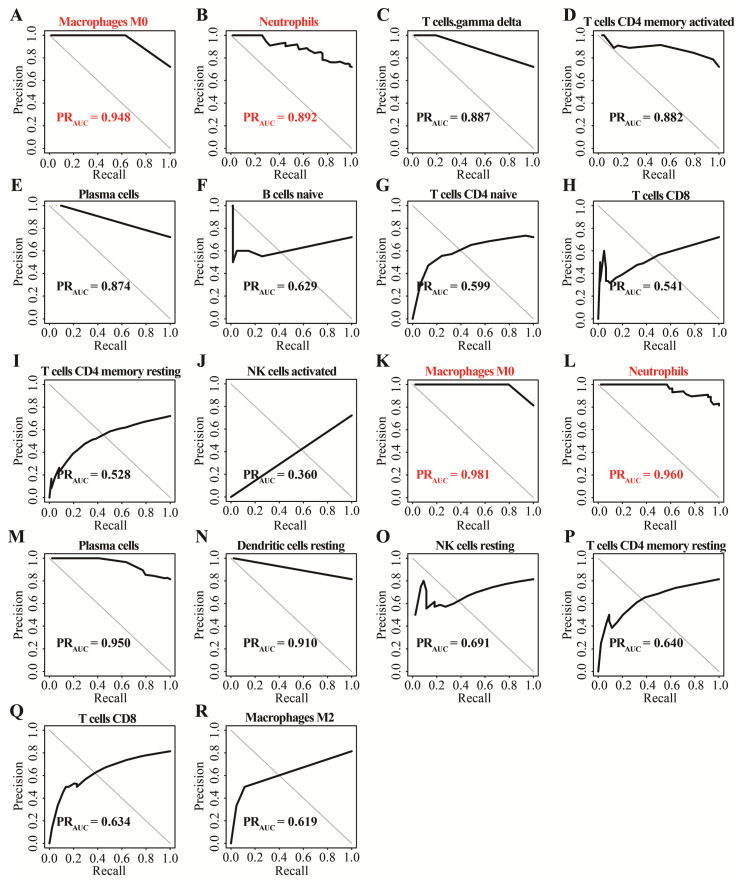
The determination of important immune cells using the precision–recall method. (**A**–**J**) The AUC of ten differentially distributed immune cells for outcomes in GSE152641. (**K**–**R**) The AUC of eight differentially distributed immune cells for outcome in GSE171110.

**Figure 4 genes-13-01602-f004:**
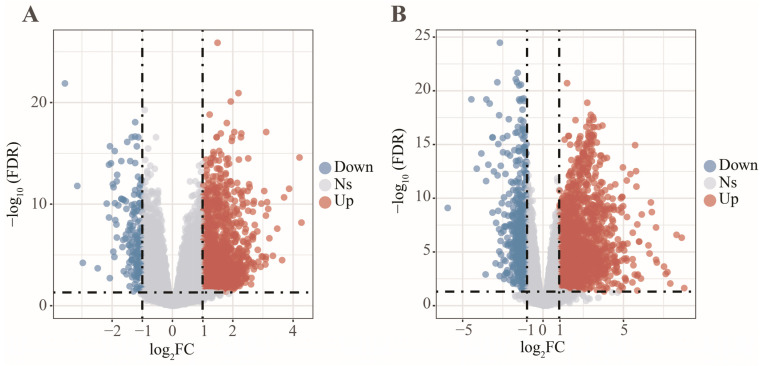
The differentially expressed analysis between COVID-19 patients and healthy controls. (**A**) The volcano plot of differentially expressed genes in GSE152641. (**B**) The volcano plot of differentially expressed genes in GSE171110.

**Figure 5 genes-13-01602-f005:**
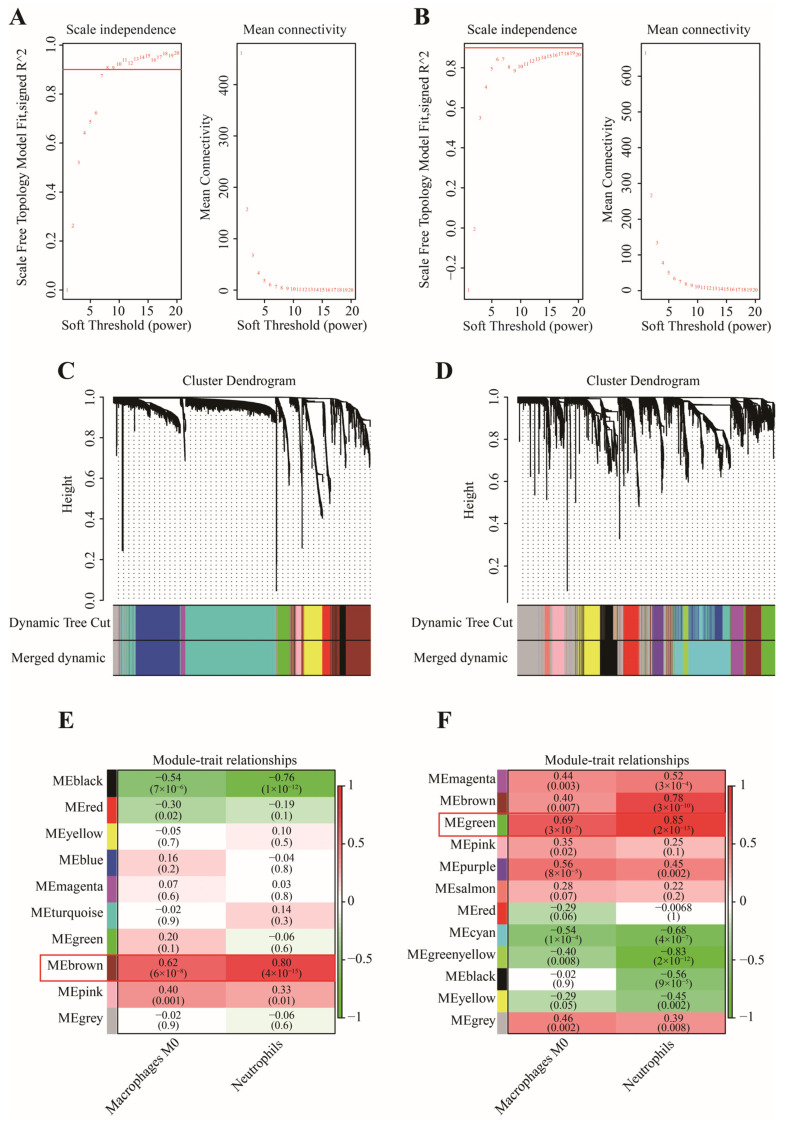
The selection of key modules associated with important immune cells. (**A**) The scale independence and mean connectivity in GSE152641. (**B**) The scale independence and mean connectivity in GSE171110. (**C**) The cluster dendrogram in GSE152641. (**D**) The cluster dendrogram in GSE171110. (**E**) The heatmap of correlation between modules and two important immune cells in GSE152641. (**F**) The heatmap of correlation between modules and two important immune cells in GSE171110.

**Figure 6 genes-13-01602-f006:**
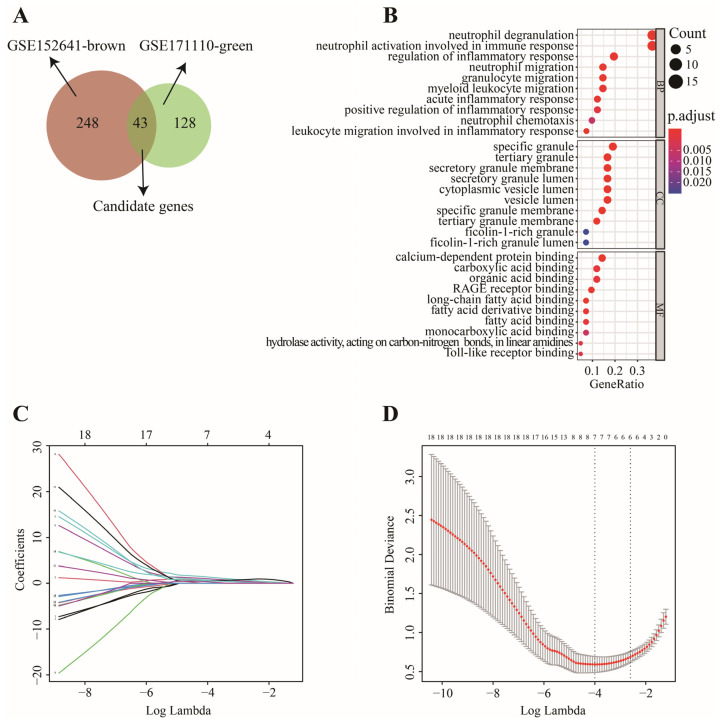
The screening process of key genes. (**A**) 43 candidate genes were intersected via Venn diagram. (**B**) “GO” enrichment analysis of 43 candidate genes. (**C**) LASSO coefficient profiles of 19 genes with confirmed importance. (**D**) Binomial deviance of genes revealed by LASSO.

**Figure 7 genes-13-01602-f007:**
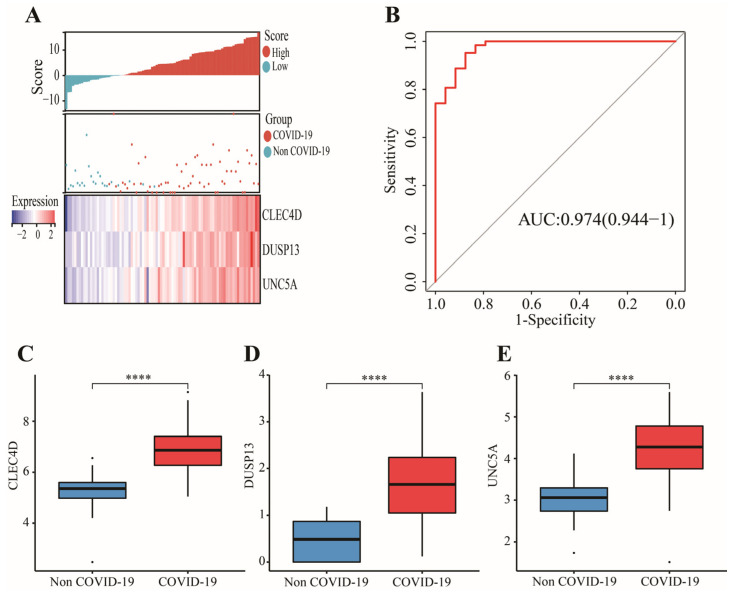
Identification of a 3-gene signature in GSE152641. (**A**) Different outcomes and expressions of CLEC4D, DUSP13, and UNC5A between high- and low-score groups. (**B**) The AUC for prediction of outcome. (**C**–**E**)The expressions of CLEC4D, DUSP13, and UNC5A between COVID-19 and non-COVID-19 groups. Data in (**C**–**E**) were analyzed by Wilcoxon test; ns, no significance; **** *p* < 0.0001.

**Figure 8 genes-13-01602-f008:**
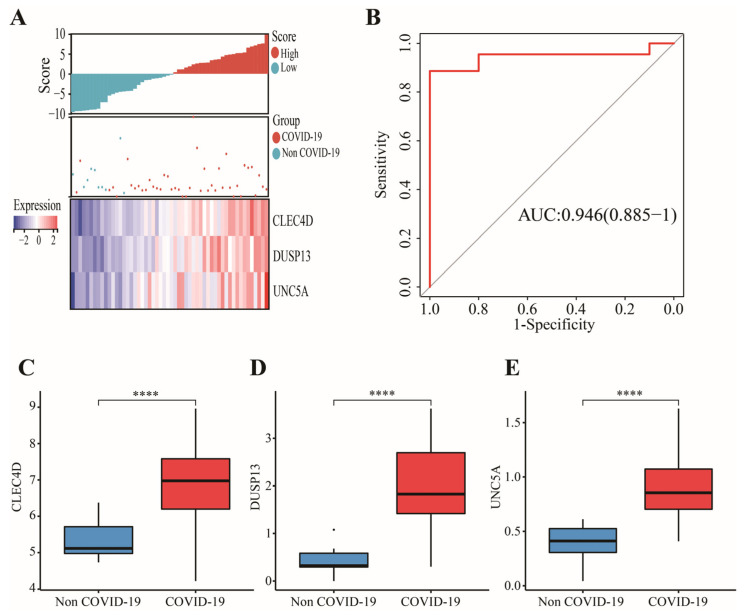
Validation of a 3-gene signature in GSE171110. (**A**) Different outcomes and expressions of CLEC4D, DUSP13, and UNC5A between high- and low-score groups. (**B**) The AUC for prediction of outcome. (**C**–**E**) The expressions of CLEC4D, DUSP13, and UNC5A between COVID-19 and non-COVID-19 groups. Data in (**C**–**E**) were analyzed by Wilcoxon test; ns, no significance; **** *p* < 0.0001.

**Table 1 genes-13-01602-t001:** The detailed results of the Boruta and LASSO algorithms.

Gene Symbol	Boruta-Decision	LASSO-Coefficient
ADM	Rejected	
ALOX5AP	Confirmed	-
ANXA3	Rejected	
ATP9A	Rejected	
BEND7	Confirmed	0.56877713
BMX	Rejected	
CA4	Confirmed	-
CD177	Confirmed	-
CKAP4	Rejected	
CLEC4D	Confirmed	1.372927226
CST7	Rejected	
DDAH2	Rejected	
DUSP13	Confirmed	0.985482029
DYSF	Rejected	
FCAR	Confirmed	-
FFAR3	Rejected	
FUT7	Rejected	
GADD45A	Tentative	
GALNT14	Rejected	
GPR84	Rejected	
GYG1	Confirmed	-
HK3	Confirmed	-
HP	Confirmed	-
IFITM10	Confirmed	0.700047504
KREMEN1	Rejected	
LRRN1	Confirmed	-
LTB4R	Confirmed	0.512153962
MCEMP1	Tentative	
MMP9	Rejected	
NR2E1	Rejected	
OPLAH	Rejected	
OSM	Confirmed	-
PADI4	Tentative	
PFKFB3	Rejected	
RAB19	Rejected	
ROPN1L	Confirmed	-
S100A12	Rejected	
S100A8	Confirmed	-
S100A9	Confirmed	-
S100P	Rejected	
UNC5A	Confirmed	0.539946778
UPP1	Confirmed	0.321891042
ZDHHC19	Rejected	

## Data Availability

The results shown here are based upon data generated by GEO (http://www.ncbi.nlm.nih.gov/geo/, accessed on 1 June 2022) and CIBERSORT (https://cibersort.stanford.edu/, accessed on 5 June 2022) databases.

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
