# Peer review of "A Novel 3-Gene Signature for Identifying COVID-19 Patients Based on Bioinformatics and Machine Learning"

_genes, 2022, doi:10.3390/genes13091602_

Round 1

Reviewer 1 Report

One major issue with the manuscript was that the validation dataset, GSE171110, was used extensively in identifying candidate immune cells and genes, which would substantially bias the results and affect the generalizability of the method.

Additionally, more detailed descriptions of the Methods are needed. For example, for the samples associated with the two GEO datasets, what were the enrollment criteria of the patients? How were the controls selected etc.?

Figure 1 is not self-explanatory. Consider adding descriptive legend and explain the abbreviations.

As the datasets are not balanced, I suggest including the precision-recall score alongside AUC.

Author Response

Dear reviewer,

Thank you for reviewing our manuscript titled “A novel 3-gene signature for identifying COVID-19 patients based on bioinformatics and machine learning.” We have carefully studied the comments and suggestions, and then revised the manuscript accordingly. The changed and added texts in the manuscript are shown in red. Also, please note that some results and figures have been changed slightly, which are different from that in the original one, because we added the one new dataset and the analysis methods. We hope that the revision could be acceptable, and that our responses adequately address the comments. Should you have any questions, please contact us without hesitation.

Q1. One major issue with the manuscript was that the validation dataset, GSE171110, was used extensively in identifying candidate immune cells and genes, which would substantially bias the results and affect the generalizability of the method.

Response: Thanks for your valuable comment. We have included another external dataset GSE196822 to validate our results. We found that a higher immune infiltration of M0 Macrophages and Neutrophils in COVID-19. CLEC4D (log2FC=1.808, FDR=0.0306), DUSP13 (log2FC=3.955, FDR=0.0002) and UNC5A (log2FC=2.350, FDR=0.0055) were up-regulated in COVID-19. Overall, the AUC in GSE196822 was 0.709, showing a predictive value of our signature. Therefore, our results can be validated in a new dataset. We have added these contents into revised manuscript (Line 75-78, Page 2; Line 89, Page 2; Line 156-157, Page 5; Line 262-271, Page 12; Line 377 Page 14; Line 513-517, Page 18).

Q2. Additionally, more detailed descriptions of the Methods are needed. For example, for the samples associated with the two GEO datasets, what were the enrollment criteria of the patients? How were the controls selected etc.?

Response: Thanks for your valuable suggestion. We have carefully described the methods in revised manuscript (Line 72-90, Page 2; Line 103-106, Page 4; Line 116-117, Page 4; Line 135-136, Page 4; Line 143-149, Page 4-5; Line 160-164, Page 5).

Q3. Figure 1 is not self-explanatory. Consider adding descriptive legend and explain the abbreviations.

Response: Thanks for your valuable suggestion. We have added descriptive legend and explainable abbreviations in Figure 1 (Line 92-94, Page 4).

Q4. As the datasets are not balanced, I suggest including the precision-recall score alongside AUC.

Response: Thanks for your valuable suggestion. We have changed the ROCAUC into PRAUC when comparing the predictive power of immune cells. The detailed analysis was shown in revised manuscript (Line 103-106, Page 4; Line 160-161, Page 5; Line 179-181, Page 5; Line 191-193, Page 6; Line 202-203, Page 7; Line 206, Page 7; Line 222-234 Page 8; Line 282-294 Page 13).

Reviewer 2 Report

- “CIBERSORT” algorithm was used to assess the fraction of immune infiltrating cells in GSE152641 and GSE171110. Key modules associated with important immune cells were selected by “WGCNA” package. “GO” enrichment analysis was used to reveal the biological function associated with COVID-19. “Boruta” algorithm was used further to screen candidate genes and “LASSO” algorithm was used for col- 13 linearity reduction.

Are the authors referring to the current study? If so the authors can include "In this work....."

The authors have to check the tenses use din the article carefully. Is it "GO” enrichment analysis was used to reveal..." or "GO” enrichment analysis is used to reveal....". Pls check this throughout the article.

- Numerical results obtained can be included in the abstract.

- Discuss about the gaps identified from existing literature that motivated the current research.

- What are the key contributions of this work?

- Some of the recent works such as the following can be discussed

Deep learning and medical image processing for coronavirus (COVID-19) pandemic: A survey

- The findings from recent state of the art can be summarized in a table.

- What is the reason behind using LASSO regression in this work when compared to other regressors? Justify.

- How are the hyperparameters chosen? Did the authors use any algorithm or is it random?

- A detailed analysis on the inferences of the authors on the results obtained has to be presented.

- What is the computational complexity of the proposed approach?

- Conclusion should include the limitations and future scope of this study.

Author Response

Dear reviewer,

Thank you for reviewing our manuscript titled “A novel 3-gene signature for identifying COVID-19 patients based on bioinformatics and machine learning.” We have carefully studied the comments and suggestions, and then revised the manuscript accordingly. The changed and added texts in the manuscript are shown in red. Also, please note that some results and figures have been changed slightly, which are different from that in the original one, because we added the one new dataset and the analysis methods. We hope that the revision could be acceptable, and that our responses adequately address the comments. Should you have any questions, please contact us without hesitation.

Q1.  “CIBERSORT” algorithm was used to assess the fraction of immune infiltrating cells in GSE152641 and GSE171110. Key modules associated with important immune cells were selected by “WGCNA” package. “GO” enrichment analysis was used to reveal the biological function associated with COVID-19. “Boruta” algorithm was used further to screen candidate genes and “LASSO” algorithm was used for collinearity reduction. Are the authors referring to the current study? If so the authors can include "In this work....."

Response: Thanks for your valuable suggestion. We have included “In this work” in revised manuscript (Line 10, Page 1).

Q2. The authors have to check the tenses used in the article carefully. Is it "GO” enrichment analysis was used to reveal..." or "GO” enrichment analysis is used to reveal....". Pls check this throughout the article.

Response: Thanks for your valuable suggestion. We have changed “GO enrichment analysis was used to reveal...” into “GO enrichment analysis is used to reveal...”. This revised manuscript has been carefully checked, especially for tenses use (Line 11, Page 1; Line 13, Page 1; Line 14, Page 1; Line 14, Page 1; Line 121, Page 4; Line 319, Page 13; Line 320, Page 13).

Q3. Numerical results obtained can be included in the abstract.

Response: Thanks for your valuable suggestion. We have included obtained numerical results in the abstract in revised manuscript (Line 16-17, Page 1; Line 21-22, Page 1). Besides, considering that the abstract required a single paragraph of about 200 words maximum and some important numeric results should be included in abstract, thus we have to remove some redundant descriptions: 1. “Brown module and green 17 module was key module in GSE152641 and GSE171110, respectively.”  2. “neutrophil degranulation, regulation of inflammatory response, RAGE receptor binding, fatty acid binding and Toll-like receptor binding activities.”

Q4. Discuss about the gaps identified from existing literature that motivated the current research.

Response: Thanks for your valuable suggestion. We have discussed the gaps identified from existing literature in revised manuscript (Line 57-64, Page 2; Line 337-348, Page 14).

Q5. What are the key contributions of this work?

Response: Thanks for your valuable suggestion. In this work, we identified two important immune cells (M0 Macrophages and Neutrophils) with high diagnostic and predictive values. Next, we screened some genes associated with the two immune cells, revealing some participatory biological functions and activities of these genes. Finally, we constructed a 3-gene signature based on machine learning. Therefore, the key contributions of this work were that we have developed a novel signature which can accurately identify COVID-19 patients from different platforms and solved the problem of few attentions on genes relevant to immune cells of COVID-19. These contributions can strengthen the potential role of two important immune cells and three candidate genes in targeting to COVID-19. (Line 354-359, Page 14). 

Q6. Some of the recent works such as the following can be discussed

Response: Thanks for your valuable suggestion. We have discussed some recent works, such as “Deep learning and medical image processing for coronavirus (COVID-19) pandemic: A survey” (Line 343-348, Page 14; Line 375, Page 14).

Q7. The findings from recent state of the art can be summarized in a table.

Response: Thanks for your valuable suggestion. We have summarized the findings from recent state of the art in revised manuscript (Line 55-57, Page 2; Line 507, Page 17).

Q8. What is the reason behind using LASSO regression in this work when compared to other regressors? Justify.

Response: Thanks for your valuable comment. Firstly, the LASSO has been commonly used for selecting the variables in bioinformatics studies [1-2]. Secondly, we compared the LASSO with ridge regression and elastic net regression using “glmnet” package. As a result, we found that the LASSO algorithm had a lowest MSE among these models, indicating LASSO was the best model. We have added these analyses into the revised manuscript (Line 143-149, Page 4-5; Line 228-231, Page 9; Line 375-376, Page 14; Line 509, Page 18).

Q9. How are the hyperparameters chosen? Did the authors use any algorithm or is it random?

Response: Thanks for your valuable suggestion. These hyperparameters were not randomly chosen in this work. The chosen hyperparameters were mainly depended on previous studies [3-13]. For example, “CIBERSORT” algorithm with 1000 permutations and selection of samples with p < 0.05 were performed in many studies [3-4]. Genes with |log2FC| > 1 and FDR < 0.05 were regarded as the differentially expressed genes when using “edgeR” package [5-6].The “WGCNA” package was used to finish following sets: 1. fitting index R2 > 0.85; 2. minimum module size =30; 3. cutHeight = 0.2 [7-9]. The minimum lambda value was obtained from 10-fold cross-validation and genes with nonzero regression coefficients were screened using LASSO algorithm [10-11]. Likewise, we ensured that the variables with p < 0.05 were retained in the final model when using stepwise Logistic regression [12-13]. As for Boruta, we considered to increase the max run times in order to ensure the stability of results though the default of max run times is 100.

Q10. A detailed analysis on the inferences of the authors on the results obtained has to be presented.

Response: Thanks for your valuable suggestion. We have added the detailed analysis on the inferences of the authors on the results obtained in revised manuscript (Line 282-294, Page 13; Line 313-317, Page 13).

Q11. What is the computational complexity of the proposed approach?

Response: Thanks for your valuable comment. As we know, computational complexity is defined as the amount of resources that is required for algorithms, including time complexity and space complexity. The time and space complexity of Boruta algorithm is respectively O (n*log(n)*f) and O (p*k). (n=samples; f=feature; k=number of trees; p=number of tree nodes) [14]. The time and space complexity of LASSO algorithm is respectively O (f2n+f3) and O (f). (n=samples; f=feature) [15-16]. As for Logistic regression, the time and space complexity is respectively O (f*n) and O (f). (n=samples; f=feature) [17].

Q12. Conclusion should include the limitations and future scope of this study.

Response: Thanks for your valuable suggestions. We have included the limitations and future scope into conclusion of revised manuscript (Line 359-373, Page 14).

References:

1. Wei, J. H.; Feng, Z. H.; Cao, Y.; Zhao, H. W.; Chen, Z. H.; Liao, B.; Wang, Q.; Han, H.; Zhang, J.; Xu, Y. Z.; et al. Predictive value of single-nucleotide polymorphism signature for recurrence in localised renal cell carcinoma: a retrospective analysis and multicentre validation study.  Oncol. 2019, 20, 591-600, doi:10.1016/S1470-2045(18)30932-X.

2. Frost, H.R.; Amos, C.I. Gene set selection via LASSO penalized regression (SLPR).  Acids. Res. 2017, 45, e114, doi:10.1093/nar/gkx291.

3. Jin, R.; Liu, C.; Zheng, S.; Wang, X.; Feng, X.; Li, H.; Sun, N.; He, J. Molecular heterogeneity of anti-PD-1/PD-L1 immunotherapy efficacy is correlated with tumor immune microenvironment in East Asian patients with non-small cell lung cancer.  Biol. Med. 2020, 17, 768-781, doi:10.20892/j.issn.2095-3941.2020.0121.

4. Han, X.; Cao, W.; Wu, L.; Liang, C. Radiomics Assessment of the Tumor Immune Microenvironment to Predict Outcomes in Breast Cancer.  Immunol. 2022, 12, 773581, doi:10.3389/fimmu.2021.773581.

5. Zhao, H.; Dang, R.; Zhu, Y.; Qu, B.; Sayyed, Y.; Wen, Y.; Liu, X.; Lin, J.; Li, L. Hub genes associated with immune cell infiltration in breast cancer, identified through bioinformatic analyses of multiple datasets.  Biol. Med. 2022, 2095-3941, doi:10.20892/j.issn.2095-3941.2021.0586.

6. Huang, R.; Li, Z.; Zhang, J.; Zeng, Z.; Zhang, J.; Li, M.; Wang, S.; Xian, S.; Xue, Y.; Chen, X.; et al. Construction of Bone Metastasis-Specific Regulation Network Based on Prognostic Stemness-Related Signatures in Breast Invasive Carcinoma.  Oncol. 2021, 10:613333, doi:10.3389/fonc.2020.613333.

7. Guo, D.; Fan, Y.; Yue, J.R.; Lin, T. A regulatory miRNA-mRNA network is associated with transplantation response in acute kidney injury.  Genomics. 2021, 15, 69, doi:10.1186/s40246-021-00363-y.

8. Guo, C.; Gao, Y.Y.; Ju, Q.Q.; Zhang, C.X.; Gong, M.; Li, Z.L. The landscape of gene co-expression modules correlating with prognostic genetic abnormalities in AML.  Transl. Med. 2021, 19, 228, doi:10.1186/s12967-021-02914-2.

9. Ding, T.; Zhang, R.; Zhang, H.; Zhou, Z.; Liu, C.; Wu, M.; Wang, H.; Dong, H.; Liu, J.; Yao, J. L. Identification of gene co-expression networks and key genes regulating flavonoid accumulation in apple (Malus × domestica) fruit skin.  Sci. 2021, 304, 110747, doi:10.1016/j.plantsci.2020.110747.

10. Rius, F. E.; Papaiz, D. D.; Azevedo, H.; Ayub, A.; Pessoa, D. O.; Oliveira, T. F.; Loureiro, A.; Andrade, F.; Fujita, A.; Reis, E. M.; et al. Genome-wide promoter methylation profiling in a cellular model of melanoma progression reveals markers of malignancy and metastasis that predict melanoma survival.  Epigenetics. 2022, 14, 68, doi:10.1186/s13148-022-01291-x.

11. Yin, X.; Wang, Z.; Wang, J.; Xu, Y.; Kong, W.; Zhang, J. Development of a novel gene signature to predict prognosis and response to PD-1 blockade in clear cell renal cell carcinoma. Oncoimmunology. 2021, 10, 1933332, doi:10.1080/2162402X.2021.1933332.

12. Gao, X.; Liu, Y.; Zou, S.; Liu, P.; Zhao, J.; Yang, C.; Liang, M.; Yang, J.  Genome-wide screening of SARS-CoV-2 infection-related genes based on the blood leukocytes sequencing data set of patients with COVID-19.  Med. Virol. 2021, 93, 5544-5554, doi:10.1002/jmv.27093.

13. Wang, H.; Zhang, Y.; Zheng, C.; Yang, S.; Chen, X.; Wang, H.; Gao, S. A 3-Gene-Based Diagnostic Signature in Alzheimer's Disease.  Neurol. 2022, 85, 6-13, doi:10.1159/000518727.

14. Hassine, K.; Erbad, A.; Hamila, R. Important Complexity Reduction of Random Forest in Multi-Classification Problem. 2019 15th International Wireless Communications and Mobile Computing Conference (IWCMC). 2019.

15. Turlach, B. Least angle regression. The Annals of Statistics, 2004, 32, 481-490.

16. Schmidt, M. Least squares optimization with L1-norm regularization. Cs542b Project Report. 2005.

17. Bulso, N.; Marsili, M.; Roudi, Y. On the Complexity of Logistic Regression Models.  Comput. 2019, 31, 1592-1623, doi:10.1162/neco_a_01207.

Round 2

Reviewer 1 Report

The authors have addressed all of my previous comments. Please fix grammatical errors in the newly added texts. 

Reviewer 2 Report

The authors have addressed all the comments. The paper can be accepted for publication.